# Characterisation and Distribution of Karaka Ōkahu Purepure Virus—A Novel Emaravirus Likely to Be Endemic to New Zealand

**DOI:** 10.3390/v13081611

**Published:** 2021-08-14

**Authors:** Lee O. Rabbidge, Arnaud G. Blouin, Kar Mun Chooi, Colleen M. Higgins, Robin M. MacDiarmid

**Affiliations:** 1The New Zealand Institute for Plant and Food Research Limited, Private Bag 92169, Auckland 1142, New Zealand; lee.rabbidge@gmail.com (L.O.R.); ablo145@aucklanduni.ac.nz (A.G.B.); Karmun.Chooi@plantandfood.co.nz (K.M.C.); 2The School of Science, Auckland University of Technology, Private Bag 92006, Auckland 1142, New Zealand; chiggins@aut.ac.nz; 3Plant Pathology Laboratory, TERRA-Gembloux Agro-Bio Tech, University of Liège, Passage des Déportés, 2, 5030 Gembloux, Belgium; 4School of Biological Sciences, The University of Auckland, Private Bag 92019, Auckland 1142, New Zealand

**Keywords:** emaravirus, karaka, *Corynocarpus laevigatus*, endemic, new-to-science

## Abstract

We report the first emaravirus on an endemic plant of Aotearoa New Zealand that is, to the best of our knowledge, the country’s first endemic virus characterised associated with an indigenous plant. The new-to-science virus was identified in the endemic karaka tree (*Corynocarpus laevigatus*), and is associated with chlorotic leaf spots, and possible feeding sites of the monophagous endemic karaka gall mite. Of the five negative-sense RNA genomic segments that were fully sequenced, four (RNA 1–4) had similarity to other emaraviruses while RNA 5 had no similarity with other viral proteins. A detection assay developed to amplify any of the five RNAs in a single assay was used to determine the distribution of the virus. The virus is widespread in the Auckland area, particularly in mature trees at Ōkahu Bay, with only occasional reports elsewhere in the North Island. Phylogenetic analysis revealed that its closest relatives are pear chlorotic leaf spot-associated virus and chrysanthemum mosaic-associated virus, which form a unique clade within the genus *Emaravirus*. Based on the genome structure, we propose this virus to be part of the family *Emaravirus*, but with less than 50% amino acid similarity to the closest relatives in the most conserved RNA 1, it clearly is a novel species. In consultation with mana whenua (indigenous Māori authority over a territory and its associated treasures), we propose the name Karaka Ōkahu purepure virus in te reo Māori (the Māori language) to reflect the tree from which it was isolated (karaka), a place where the virus is prevalent (Ōkahu), and the spotted symptom (purepure, pronounced pooray pooray) that this endemic virus appears to cause.

## 1. Introduction

Karaka (*Corynocarpus laevigatus*) is an endemic tree occurring naturally in the coastal and lowland forests of Aotearoa New Zealand’s North Island [1]. Based on fossilised wood and pollen layers within soil, karaka is recorded as having been present in New Zealand for more than five million years [2], although the frost-sensitive tree likely only survived in northern parts of the country during glacial periods [3]. It is the only species of the family Corynocarpaceae indigenous to New Zealand, with other members being in New Caledonia, Vanuatu, Papua New Guinea and Australia (Queensland and northern New South Wales) [4] Karaka means ‘orange’ as an adjective in the Māori language to describe the color of its fleshy fruit, while the Latin genus name is taken from its ‘club seed’, and the Latin species name describes its ‘smooth’ fruit [3,5]. Karaka trees grow to ~20 m in height, have thick, elliptic and glossy leaves up to 30 cm in length, and produce large fruit drupes up to 40 mm in length [6]. The fruit, especially the inner kernel, has historical and cultural significance to Māori as an important carbohydrate source eaten during winter when other food was in short supply. The kernel contains a glucoside toxin, karakin, that can cause spasmodic pains and permanent partial paralysis if ingested [7,8]. However, preparation through steaming and soaking in running streams allows the kernel to be eaten. Historically, karaka provided Māori with a staple carbohydrate source [4]. Traditionally, the kernels held importance in Māori ceremonies, funerals and formal exchanges between iwi (tribes) [4]. Flavoursome flesh of the karaka drupe can be eaten directly; the leaves have traditionally been used as a poultice [3]. The much-prized karaka trees were planted and cultivated near Māori occupation sites around New Zealand, which explains how the trees were found in the South Island of New Zealand at the time of first European contact [5,9]. The tree was also spread by Māori to offshore islands such as the Kermadec and Chatham Islands, where it formed a valued food resource for the indigenous people and an important forest habitat for indigenous birds [9]. The Moriori people of Rēkohu Chatham Islands (New Zealand) view the karaka tree or—kopi in their language—as sacred, and it is considered by them as significant to their culture and history [9].

Karaka trees in contemporary New Zealand are geographically widespread and comprise a mixture of naturally sown and hand-planted trees. The tree was likely an important food supply for now extinct indigenous birds of New Zealand. However, the kereru, an extant native pigeon, eats the fruit and today acts as its sole natural dispersal agent [5]. Karaka are desired as ornamental trees in public parks and private gardens, though have been described as weedy as they germinate prolifically and grow rapidly [3]. The tree is available at commercial nurseries throughout New Zealand.

The karaka has very few known pests and pathogens. Only one report of fungi causing symptoms was found [10], while cucumber mosaic virus and ribgrass mosaic virus are the only two viruses reported in karaka before this report [11,12]. No arthropod pests are recorded on the karaka, with the exception of the Eriophyoidea mite *Aculus corynocarpi* [13]. Eriophyid mites are well-known vectors of emaraviruses and the relationship between each emaravirus with a single plant host and, where known, with a specific monophagous mite [14]. Emaraviruses have been detected in eriophyid mites, with transmission by mites confirmed for some [15,16,17,18,19]. A. corynocarpi is endemic to Aotearoa New Zealand and relies entirely on the karaka tree as a food source and to complete its lifecycle [20]. The eriophyid mite feeds on sap from young leaves and inflorescences and major infestations can be detected through the resulting and characteristic leaf distortion or discoloration, floral bud scaring or foliar blackening. Although the mite can feed on either the abaxial (underside) or adaxial (upper surface) leaf laminar, it breeds on the underside of the karaka leaf. It is suspected that the karaka gall mite is the vector of a newly discovered emaravirus in karaka described hereafter. 

The Emaravirus genus was established in 2012 and is characterised by each member species having a segmented, single-stranded, negative-sense RNA genome and a spherical virion of 80–100 nm [21]. The type species of this genus is European mountain ash ringspot-associated virus (EMARaV) and a characteristic feature of this genus is the appearance of double-membraned bodies in the cytoplasm of infected plants [22]. Emaravirus genomes comprise four to nine negative-sense RNA strands with termini of ~13 nucleotides (nt) that share high intraspecies homology [23,24]. Distinct emaravirus species are characterised by RNA 1, 2 and 3 encoded proteins differing by >25% [25]. To date, the only emaravirus reported in New Zealand is the exotic fig mosaic virus and other viruses identified in indigenous plants of Aotearoa New Zealand are exotic [12,26,27]. 

In 2015, a karaka tree bearing leaves with chlorotic spots was observed at the New Zealand Institute for Plant and Food Research (Plant & Food Research) Mount Albert Research Campus (MARC) in Auckland. This study looked to first identify the potential causal agent for the chlorotic spots using high throughput sequencing (HTS) approaches, to understand the genetic relationship between the novel emaravirus and other similar genus members, then to correlate the presence of symptoms and the virus, and finally to assess the distribution of the virus within a wider Auckland region. Collectively, the information was then used to endow the most appropriate name, in consultation with mana whenua (indigenous Māori authority over a territory and its associated treasures). The tentative name of the virus is Karaka Ōkahu purepure emaravirus (KŌPV) and is used hereafter.

## 2. Methods and Materials

### 2.1. Virus Discovery and Genome Characterisation

#### 2.1.1. Plant Material, Nucleic Acid Isolation, and Sequencing Platform

Symptomatic leaf samples were collected in 2015 from a single tree at the MARC site of Plant & Food Research in Auckland, New Zealand. From symptomatic leaves, double-stranded RNA (dsRNA) was isolated and sequenced as described previously [26]. In brief, the dsRNA was enriched using specific antibodies and library preparation on the combined samples was performed with TruSeq Nano DNA Library Prep Kit. Illumina; sequencing was done by Macrogen Korea on HiSeq 2000 (Illumina) using a partial run with 100 bp paired end (pe). Total RNA was extracted using a modified cetyltrimethyl ammonium bromide (CTAB) based method [28] and the samples were sequenced on MiSeq (2 × 300 paired ends) by New Zealand Genomics Ltd. at the University of Auckland Genomic facilities. The small RNA (sRNA) was extracted with mirVana miRNA Isolation Kit (Ambion, Thermo Fisher Scientific). The library was prepared and sequenced on an Illumina HiSeq by BGI Group (https://en.genomics.cn/ accessed on 25 May 2021) as described by Blouin et al. [26].

The sequencing data were assembled de novo using Trinity v. v2.3.2 [29] and the contigs were identified as emaravirus by BlastX analysis performed on the NCBI protein database. The sRNA reads were mapped onto the emaravirus-identified contigs using Bowtie2 [30]. The sequence of each RNA segment was completed using 3′ rapid amplification of cDNA ends (RACE) followed by comparison with sRNA data. Essentially, the negative-sense genomic RNA and positive-sense transcript RNAs within total RNA isolated from symptomatic leaves were polyadenylated at their 3′ ends using polyA polymerase (New England Biolabs). The 3′ end of each viral RNA was amplified by one-step reverse transcription polymerase chain reaction (RT-PCR) with a gene-specific primer paired with a poly-T-SP6 primer (PV1) (see Appendix A for primer sequences) using the Invitrogen Superscript™ RT-PCR with Platinum™ Taq DNA polymerase kit (Thermo Fisher Scientific Waltham, MA, USA). The RT-PCR conditions were 1 cycle of cDNA synthesis for 30 min at 50 °C, 1 cycle of initial denaturation for 2 min at 98 °C, 35 cycles of PCR amplification with denaturation for 10 s at 98 °C, annealing for 30 s at 55 °C, extension for 50 s at 72 °C, followed by a final extension for 7 min at 72 °C. A second PCR was performed using CloneAmp HiFi PCR Premix (Takara Bio Inc, Mountain View, CA, USA.) with each gene-specific primer and the SP6 primer using 1 µL from the previous RT-PCR. The PCR conditions were 1 cycle of initial denaturation for 30 s at 98 °C, 35 cycles of PCR amplification with denaturation for 10 s at 98 °C, annealing for 30 s at 52 °C, and extension for 50 s at 72 °C, followed by a final extension for 7 min at 72 °C. PCR products were gel purified using the Invitrogen PureLink PCR Purification Kit (Thermo Fisher Scientific) as per the manufacturer’s instructions and Sanger sequenced (Macrogen, Seoul, Korea). 

#### 2.1.2. Genome and Protein Analysis

To confirm the terminal nucleotides at each end of each RNA segment, sequences obtained from the RACE analysis of each segment were analysed using Geneious v6.1.8 software (https://www.geneious.com accessed on 25 May 2021, Biomatters, Auckland, New Zealand) by pairwise alignment to their respective RNA sequence previously generated by Illumina sequencing. To confirm the tentatively completed genome was correct, sRNA from the NGS data was mapped to the completed genome sequence for each segment by reference assembly using Geneious v6.1.8. Any discrepancies between the two sequences were analysed and the genome sequence amended based on the number of reads in the original NGS sequence and the number reads in the sRNA assembly giving the final genome (GenBank #MZ391827–MZ391831). 

To identify similar nucleotide (nt) sequences, assembled contigs were used as query sequences using BLASTn to search the GenBank database within the National Center for Biotechnology Information website (NCBI, www.ncbi.nlm.nih.gov accessed on 25 May 2021) [31]. BLASTx searches were carried out in NCBI to identify similar proteins. Putative domains within the predicted aa sequences encoded by each RNA segment were identified from the Conserved Domain Database (NCBI). Predicted aa sequences were further characterised as follows: molecular weights were predicted using The Protein Molecular Weight tool (https://www.bioinformatics.org/sms/prot_mw.html accessed on 25 May 2021); putative N-glycosylation and O-glycosylation sites were identified using NetNGlyc 1.0 NetOGlyc 4.0, respectively [32,33]; signal peptides were predicted using SignalP 5.0 [34]; transmembrane regions were predicted using Tmpred [35]; the presence of N-terminal pre-sequences that may suggest involvement in a secretory pathway was predicted using TargetP 2.0 [36]; COILS was used to predict the presence of any coiled coils [37].

The virus sequences and their predicted protein products were compared with genome and protein data of all known emaraviruses. Percent pairwise identities for the protein products of RNA 1, RNA 2 and RNA 3 are provided in the Appendix A (Appendix A). 

#### 2.1.3. Phylogenetic Analysis

The relationship between KŌPV and other emaraviruses was investigated using phylogenetic analysis. The P1 aa sequences (polymerase or L) were used for comparison with KŌPV (listed in Appendix A). Sequences were aligned using the MUSCLE algorithm and maximum likelihood analyses were carried out in MEGA X [38] with Bunyamwera virus (NP_047211), tomato spotted wilt virus (BAA00955) and impatiens necrotic spot virus (NP_619710) used as the outgroups. The evolutionary model applied was LG + G + I + F [39] with 1000 bootstraps [40].

### 2.2. Survey for KŌPV

#### 2.2.1. Geospatial Location for Plant Collection 

Surveying and collection of samples in Auckland were performed using a 2 km × 2 km grid created in QGIS [41]. The survey was carried out predominantly on foot by one experienced person using only first-person observations. Surveying efforts concentrated on delimiting the distribution of karaka trees with at least one representative per grid square within the immediate central Auckland area. This approach focused on distribution rather than abundance of karaka trees in Auckland, Aotearoa New Zealand. A limited number of trees were sampled outside of the central Auckland area (*n* = 12).

#### 2.2.2. Plant Material and RNA Isolation

In total, 115 symptomatic and 143 asymptomatic trees were surveyed and sampled. From the symptomatic trees, a symptomatic leaf and an asymptomatic leaf were collected; from the asymptomatic trees, only an asymptomatic leaf could be collected. Total RNA was extracted from leaf samples using the CTAB method [28], and a multiplex RT-PCR diagnostic test was performed. 

#### 2.2.3. Diagnostic Method

Karaka RNA was assessed using the internal primers VvNAD5 developed by Chooi et al. [42] to test integrity of RNA for the RT-PCR. One-step RT-PCR primers specific to each segment of KŌPV were tested on leaf material from MARC and the amplicons sequenced to confirm positive results were of the correct sequence (Primers listed in Appendix A). A one-step RT-PCR assay was developed that combined primers for the five segments together with amplicons appearing together as a single band. Generic primers developed by Elbeaino et al. [43] to detect all known emaraviruses at the time of publication were tested on karaka RNA. 

## 3. Results

### 3.1. Virus Discovery and Genome Analysis

Virus-like symptoms (distinct pale green chlorotic spots; Figure 1A) on several karaka leaves were identified on a single tree at the Plant & Food Research MARC site in Auckland (2015). From the initial sequencing analysis of symptomatic leaves, (from the dsRNA enrichment), 249,692 reads were obtained, and from the BLASTX analysis of de novo assemblies against a viral protein database, two contigs of 945 and 772 nt showed low homology to the movement and coat proteins encoded by members of emaravirus, respectively. The data obtained from total RNA sequencing (7,492,776 reads) were assembled with SPAdes and the contigs compared with a viral protein database using BLASTX. Five contigs showed homology to emaravirus sequences, with lengths of 7333 nt, 1927 nt, 1751 nt, 1601 nt and 1506 nt. In total 8474 reads mapped the five contigs which represents an average coverage of 180× (individual segment coverage is presented in Appendix A). The length of each RNA genome segment and termini sequence was subsequently confirmed by 3′ RACE (Figure 2). The sRNA sequencing yielded a total of 29,013,508 reads, from which 2,457,123 reads fully mapped to the five contigs using Geneious (Medium-Low sensitivity) with a fold coverage that ranged between 1048× for RNA 2 to 11,493× for RNA 3 (3660× fold coverage on average, individual segment cover ge is presented in Appendix A).

KŌPV consists of five RNA segments, supported by the total RNA sequencing data and the high coverage and depth of the sRNA data. The genome structure of KŌPV shares similarity with other emaraviruses (Figure 2). In brief, the five RNA segments each encode a single protein. Importantly, the first 13 nucleotides at the termini of each RNA segment are highly conserved between the five segments, with the first 7 nt (ATGAGTG) being invariable, a characteristic possessed by all emaraviruses published to date. 

The KŌPV RNA 1 is 7141 nt in length with an open reading frame (ORF) found between AUG64-66 and UUA7024-7026 (Figure 2) that encodes a putative 271.13 kDa protein (P1) of 2320 aa, and shares 46% aa identity with chrysanthemum mosaic-associated virus (ChMaV) (Appendix A). The P1 protein has a conserved domain from aa 653–1395 with an E-value of 1.22 × 10^−34^ that belongs to the bunyavirus RNA-dependent RNA polymerase (RdRp) superfamily (Accession cl20265). This indicates that P1 is the RdRp [31]. P1 is the only KŌPV protein that contains recognised domains from the Conserved Domain Database [44]. Motif A (DASKWS) is highly conserved across all emaravirus RdRps and is found at aa 1128–1133 [22,45,46]. Motif B (GNLNRLSS) is found at aa 1213–1221; this sequence matches the conserved QGNXNXXSS sequence [45]. Motif C (SDD) and E (EFLST) are also highly conserved across all emaraviruses and are found at aa 1254–1256 and 1311–1315, respectively, Motif D (KK), lies between Motifs C and E of P1 [45,47]. In KŌPV, there appear to be two Motifs D at aa 1266–1267 and downstream at aa 1301–1302. P1 shares the N-terminus endonuclease domain with other emaraviruses and is located at RHD106-108X35PD145-146X12EVK158-160. P1 also has the conserved premotif A sequence KDQRTYNDREIYTGNKEAR positioned at aa 1050–1068 with the closest similarity to the premotif A sequences of jujube yellow mottle-associated virus (JYMaV), raspberry leaf blotch virus (RLBV) and High Plains wheat mosaic virus (HPWMoV) [45,48,49,50].

RNA 2 is 1943 nt in length and contains a large ORF between AUG59-61 and UAG1856-1858 and is predicted to encode a glycoprotein (P2) of 599 aa, predicted MW = 68.23 kDa. The protein shares the highest aa identity with chrysanthemum mosaic-associated virus (ChMaV) (31.9%). Unexpectedly, the KŌPV P2 does not appear to contain any N-glycosylation or O-glycosylation sites; emaraviruses have been previously reported to contain between four and six N-glycosylation sites [18,46,51]. Moreover, the tetrapeptide sequence, ADDN, predicted in other emaravirus glycoproteins to cleave the protein into two smaller proteins, is not present in P2 [45]. A potential signal peptide (probability of 0.9692) with a cleavage site between TTS17-19 and KY20-21 was identified by SignalP 5.0. The likelihood of the cleavage site was weakly supported with a probability of 0.3522. The signal peptide was further supported by the TargetP 1.1 analysis indicating a signal peptide within the first 20 aa. This signal peptide may be involved in targeting P2 to the endoplasmic reticulum, based on similar analysis of Rose rosette virus [51]. P2 is predicted to contain five transmembrane helices, which is more than the two to four helices reported to be present in other emaraviruses [22,45,51]. 

RNA 3 is 1479 nt in length and contains an ORF between UUG42-44 and UAA963-965 that is predicted to encode a nucleocapsid protein (P3) of 307 aa, predicted MW = 34.49 kDa. The protein has 37.6% aa identity with ChMaV and contains three conserved aa regions: A (NKFVMSSNR119-127), B (NRLA173-176) and C (GVEN194-197) (Figure 2). Region A of KŌPV is the most dissimilar to the RNA 3 coding regions of other emaraviruses, whereas region B is similar to that found in all other emaraviruses [46]. Region C of KŌPV is the only emaravirus with a valine (V) residue in the second position. 

RNA 4 is 1518 nt in length with an ORF between AUG103-105 and UAA1054-1056 that is predicted to encode a movement protein (P4) of 317 aa, predicted MW = 36.20 kDa. A signal peptide was detected between aa positions 1 and 18 using SignalP 5.0 and it is in a similar position to a signal peptide Actinidia chlorotic ringspot-associated virus (AcCRaV) [47]. Although the P4 encoded by KŌPV does not appear to contain any conserved domains, particularly the emaravirus 30 kDa movement superfamily domain, the putative movement protein does have similarity with other emaravirus P4 sequences. It shows 58% aa identity with pear chlorotic leaf spot-associated virus (PCLSaV) and 58% identity with ChMaV; interestingly, these viruses also do not appear to have the 30 kDa movement superfamily domain.

RNA 5 is 1576 nt in length with an ORF between UUG87-89 and UAA894-896 that encodes a protein with unknown function (P5) of 269 aa, predicted MW = 30.30 kDa. The aa sequence appears to share no similarity with any other viral protein nor has any conserved domains. The aa sequence has no coiled-coil domains, signal peptides for a secretory pathway or cleavage sites, and shows a poor likelihood for forming transmembrane helices. Four candidate N-glycosylation sites and ten potential O-glycosylation sites were identified using NetNGlyc 1.0 and NetOGlyc 4.0, respectively. 

### 3.2. Phylogenetic Analysis

Phylogenetic analysis of the P1 protein was undertaken to determine the relationship between KŌPV and other emaraviruses (Figure 3). Amino acid sequence analysis of the putative RdRp of known emaraviruses showed the emaraviruses comprise three major clades (clades I, II and III), with the largest clade (clade I) split into three subclades (subclades a, b and c). The observed branching pattern conforms to previous reports [52,53,54,55]. The earliest identified emaraviruses belong to Clade I, while more recent discoveries have expanded our understanding of emaravirus evolution with the identification of clades II and III. Clades II and III are independent lineages of clade I; KŌPV belongs to clade II, along with ChMaV and PCLSaV.

Amino acid identities for proteins encoded by RNAs 1, 2 and 3 are provided in Appendix A. For P1 of KŌPV, the RdRp, the closest match is that of ChMaV at 46% amino acid sequence identity. For the glycoprotein (P2) encoded by RNA 2 and the nucleocapsid (P3) encoded by RNA 3, the ChMaV genome has the highest sequence identity at 31.9% and 37.6%, respectively. 

### 3.3. Detection and Distribution

The distribution of KŌPV in central Auckland was investigated by surveying and sampling symptomatic and asymptomatic trees. A subset of RNA samples from 40 asymptomatic leaves from symptomatic karaka trees and 40 asymptomatic leaves from asymptomatic trees were tested with the internal primers VvNAD5 which validated the karaka RNA isolation method and the integrity of RNA for the RT-PCR (data not shown). From 258 trees sampled, 44.6% showed symptoms (Figure 4). The location of symptomatic trees across Auckland did not appear to be clustered, but rather trees were evenly distributed throughout the survey area and displayed the same chlorotic spots observed at the initial Auckland site (Figure 5). Using RNA from symptomatic leaves the multiplex RT-PCR diagnostic test gave stronger amplification than the generic primers of Elbeaino et al. [43] (data not shown) therefore the multiplex was used subsequently. A positive correlation between the presence of symptoms and the presence of KŌPV was established by multiplex RT-PCR detection of the virus in symptomatic and asymptomatic leaves (Figure 1B and Figure 5). Of the 115 symptomatic trees, 101 (87.8%) tested positive for KŌPV using symptomatic leaves, whereas only 7 (6.1%) tested positive based on testing of asymptomatic leaves (Figure 5). Of the 143 asymptomatic trees, 4 (2.3%) tested positive for KŌPV. From these results (Figure 5), sensitivity of the visual symptoms as an indicator of virus infection as confirmed by RT-PCR (percentage of symptomatic leaf from positive RT-PCR) was determined to be 90% and the specificity (percentage of asymptomatic leaf from negative RT-PCR) was determined to be 95%. The positive predictive value (percentage positive RT-PCR from symptomatic leaf) was 87.8% and the negative predictive value (percentage negative RT-PCR from asymptomatic leaf) was 96%.

## 4. Discussion

Virus-like symptoms were identified on several karaka leaves at the Plant & Food Research MARC site in Auckland in 2015. To our knowledge, these symptoms have not been previously described in literature or transcripts of Māori oral histories. Cucumber mosaic virus was reported to infect karaka in Christchurch (South Island) showing mosaic and concentric ring markings [11]. Additionally, no leaf deformation matching the chlorotic spot symptoms was found when examining digital photographs of karaka leaf collections maintained by the Auckland War Memorial Museum. To identify the potential causal agent, non-targeted approaches were used to discover the presence of a negative-sense, segmented RNA genome that appeared to belong to the recently established Emaravirus genus. Here, we discuss the genomic relationship between the novel virus and other emaraviruses and the geographic distribution of infected trees, and how the current collective knowledge indicates KŌPV as the first description of a likely endemic virus of Aotearoa New Zealand on an endemic plant of the country. 

### 4.1. Genome Analysis

The complete KŌPV genome sequence shares the fundamental prerequisites for membership of the Emaravirus genus. It has at least five segmented, negative-sense RNA segments (RNA 1–5) of which RNA 1 to 4 appear to encode an RdRp, glycoprotein, nucleocapsid and movement protein, respectively. These identifications are based on predicted protein size, and the presence of motifs and features reported previously for emaraviruses [22,46]. While it is possible that more RNA segments exist for KŌPV, as described for other emaraviruses [56,57], this is unlikely because of the significant sequence depth obtained from the Illumina sequencing of the mRNA population, and no additional RNA segments being revealed by sRNA sequencing (Appendix A). The termini of each RNA segment share sequence similarity between the RNA segments of KŌPV and segments of other emaraviruses [46]. KŌPV is, however, genetically distinct within the genus, with relatively low pairwise identity with other emaraviruses as well as membership within a clade that represents an independent lineage within the genus. Species demarcation of emaraviruses requires greater than 25% sequence divergence for RNAs 1, 2 and 3 [58]; the sequence comparisons between KŌPV and the other known emaraviruses indicate that KŌPV is a new species of emaravirus. 

KŌPV has unique genome features. The predicted glycoprotein encoded by RNA 2 does not appear to have a glycosylation site, nor the expected cleavage site. The protein predicted from RNA 4 (P4) is considered to be the putative movement protein since it shows overall similarity to the movement proteins of other emaraviruses, despite not having the expected conserved emaravirus 30 kDa movement superfamily domain [45,47]. Interestingly, KŌPV P4 is not unique in this regard, as other emaraviruses such as ChMaV and pear chlorotic leaf spot-associated virus also lack this domain. This lack of a movement protein domain may indicate a reduced functional capacity for virus movement within its karaka host in accordance with a lack of evidence for systemic infection of the karaka tree. The predicted protein encoded by KŌPV RNA 5 appears to have no similarity to any other virus protein, and therefore its function is unknown. The closest potential orthologue is the aa sequence of Pigeonpea sterility mosaic virus P6 with which KŌPV has a 17.6% pairwise identity match. Another potential orthologue is HPWMoV P7, which is of a similar length and has a pairwise identity of 16.7%., HPWMoV P7 and P8 both encode proteins involved in suppression of the anti-viral RNA interference (RNAi) mechanism [59,60,61]. Based on these studies, it is appropriate to infer that the KŌPV P5 may be involved in RNAi suppression, although direct evidence is required to demonstrate KŌPV P5 RNAi suppression functionality.

### 4.2. Phylogeny

Designation of the Emaravirus genus into phylogroups is under constant review as new emaraviruses are discovered. The clade to which KŌPV belongs has only recently been identified [52,62], highlighting that emaraviruses are more diverse and their evolutionary history more complicated than previously understood. Other members of this recently identified clade include ChMaV and PCLSaV. The maximum likelihood analysis demonstrated that this clade has a different lineage to that of other emaraviruses. We propose a new naming for the phylogroups within the Emaravirus genus that builds on previous naming—the three separate lineages to be named groups I, II and III, and the largest group (I) to be split into three subgroups a, b and c (Figure 3). As more emaraviruses are discovered, it is likely this proposed naming will need to be reviewed; however, the current proposition recognises the apparently complicated evolutionary histories of the various emaraviruses discovered to date.

### 4.3. Diagnostics and Distribution

Karaka trees are widely found within Auckland, with evidence that symptomatic and asymptomatic trees appear to be evenly distributed. No obvious point of origin for the virus was observed. However, the origin could have been overlooked in this study, as only a subset of karaka trees were sampled. Our survey and sampling protocols prioritised the distribution of trees within a geographic region over the abundance of trees; therefore, regions that could potentially contain a denser abundance of symptomatic trees were accessed during this study. In addition, we avoided private property and sacred places (for example, the cemetery (urupā) at Orakei Beach, Auckland) because permission to enter had not been sought. However, we did observe apparent evidence of symptomatic karaka trees in some locations. A future study focused on determining the abundance of symptomatic karaka in Auckland could provide greater insight as to a potential point of origin for the KŌPV. Indeed, future surveys should include assessments of karaka throughout New Zealand, as detailed by Atherton et al. [63]. Further, analysis of the sequence variability among KOPV isolates may help determine if the emergence of this virus is recent or ancient. 

Another scenario for the even distribution of KŌPV observed in Auckland is that KŌPV initially spread (perhaps by eriophyid mites) through a nursery, followed by an artificial spread by human activity to the rest of Auckland and potentially wider New Zealand through planting of the infected trees. In this study, we could not determine with confidence whether a sampled tree was the result of natural seed dispersal or was planted from nursery material. Future surveys may consider investigating methods of differentiating naturally sown trees from purposefully planted trees to assist with determining the natural viral spread. Atherton et al. [63] suggested using specific genetic markers to determine the dispersal of karaka, which may help to differentiate seed dispersal from human-mediated activity. 

The number of symptomatic leaves present varied greatly from one karaka tree to another. Some trees displayed a few symptomatic leaves only whereas on other trees symptoms were widely observed throughout the canopy. Karaka trees were preferentially surveyed in public spaces and then visually assessed for symptoms; however, trees with few or rudimentary foliar symptoms may have been falsely recorded as asymptomatic. The survey was carried out predominantly on foot by the same person and, as a result, tall trees could not be completely assessed without a ladder or cherry picker, for example. Use of unmanned aerial vehicles may be trialed in future to more comprehensively assess individual trees and to reach trees in more remote or inaccessible locations. 

Previously, emaraviruses have been found in association with eriophyid mites, including pear chlorotic leaf spot-associated virus which Kubota et al. [64] detected within a suspected eriophyid mite vector, and in the case of Rose rosette virus, where Di Bello et al. [57] demonstrated the ability of eriophyid mites to transmit the virus. Similar studies with KŌPV and the Karaka gall mite would clarify their relationship, including whether the primary host of KŌPV is the mite with incidental infection of karaka that may contribute towards its mite to mite transmission. Such a novel scenario may even be common among eriophyid mites and emaraviruses, which comprise a double lipid envelope—a characteristic that is a requirement for infection of insects rather than plants. For instance, rhabdoviruses are likely viruses of insects that have gained a movement protein required for cell-to-cell and systemic infection of plants [65,66]. 

Perhaps KŌPV represents an example of an emaravirus that has not yet gained a movement protein to confer entry into the phloem companion cells to achieve cell-to-cell movement and assist with systemic infection. Karaka could be the plant reservoir for an eriophyid mite virus that depends on uninfected mites feeding on the viruliferous chlorotic spots for transmission to its new mite host. This transmission path that severely limits the geographic spread of KŌPV would suggest that KŌPV is also not vertically transmitted by seed nor pollen, both of which require systemic infection, or to putative KŌPV host karaka gall mite offspring. An alternative hypothesis may be a more recent arrival and evolution at or near Ōkahu Bay, which is adjacent to a major port that is currently operational and historically has been home for Māori for ~900 years. Further research on the associations between KŌPV, karaka and karaka gall mite will deepen our understanding of the evolution of the virus and provide further insights into its associations within Aotearoa New Zealand.

### 4.4. Cultural Importance and Potential to be Endemic

The novel emaravirus found in karaka has tentatively been named Karaka Ōkahu purepure emaravirus (KŌPV). Parts of the name are in the Māori language to reflect the importance of the endemic host plant to Māori. Many existing karaka groves in New Zealand have cultural significance as a historical food source and are therefore taonga, meaning “treasured”, creating an obligation for conservation of the trees in these groves [3]. Ōkahu in the name refers to the location of Ōkahu Bay in Auckland, where there is an abundance of old, naturally sown karaka trees that exhibit KŌPV symptoms. Purepure in the Māori language means spotted, which refers to the symptoms, and the emaravirus suffix adheres to the current naming convention of emaraviruses [57].

This present research has raised questions about KŌPV, such as where it came from and whether it is endemic to New Zealand. The phylogenetic analysis indicated that KŌPV shares a common ancestor with the rest of the emaraviruses but appears genetically distinct. A molecular clock study could be performed to determine the evolutionary age of the virus, as has been performed before for many plant RNA viruses [67,68]. This, coupled with research by Atherton et al. [62] on determining the evolutionary history of the karaka tree, and perhaps the karaka gall mite, could give greater insight into the origins of the virus. If KŌPV is endemic to New Zealand, it has likely co-evolved with karaka and its dependent mite. If this is the case, then historic herbarium samples and photographs could be searched for signs of the KŌPV symptoms or mites. We have undertaken this task to a limited degree by searching the Auckland War Memorial Museum’s photographic collections of karaka. Symptoms could not be found, although that may be due to deterioration of chlorosis during the storage of the samples but also due to collectors selecting “unblemished” samples for preservation. Furthermore, there appears to be no mention of the symptoms in any literature, perhaps due to the limited distribution of symptomatic trees. Research on karaka is limited; therefore, it would be prudent to explore further with tangata whenua (the local Māori people) of areas where karaka has been cultivated to determine whether the chlorotic spot symptoms had ever been observed and recorded in their written or oral history. Discussion with Kaumātua Grant Hawke (personal communication), who has eaten karaka from his favourite trees at Ōkahu Bay for over 70 years, revealed that he was not previously aware of the chlorotic symptoms. Importantly, not all leaves of the tree are affected; we have shown that symptoms are associated with the presence of KŌPV but systemic infection was not detected. This lack of systemic infection could account for the symptoms not being noticed previously and also may limit negative impacts on the growth and productivity of infected karaka. If KŌPV is endemic to New Zealand, to the best of our knowledge, it will be the first fully sequenced plant-associated virus endemic to the country and therefore important to study further to understand the virus and its ecology in this country. 

## 5. Conclusions

Karaka Ōkahu purepure virus is a novel RNA virus present within karaka (*Corynocarpus laevigatus*) that belongs to the Emaravirus genus. It is present in Auckland and in parts of the North Island. A te reo Māori name was chosen to reflect the importance of the endemic host plant to Māori. 

## Figures and Tables

**Figure 1 viruses-13-01611-f001:**
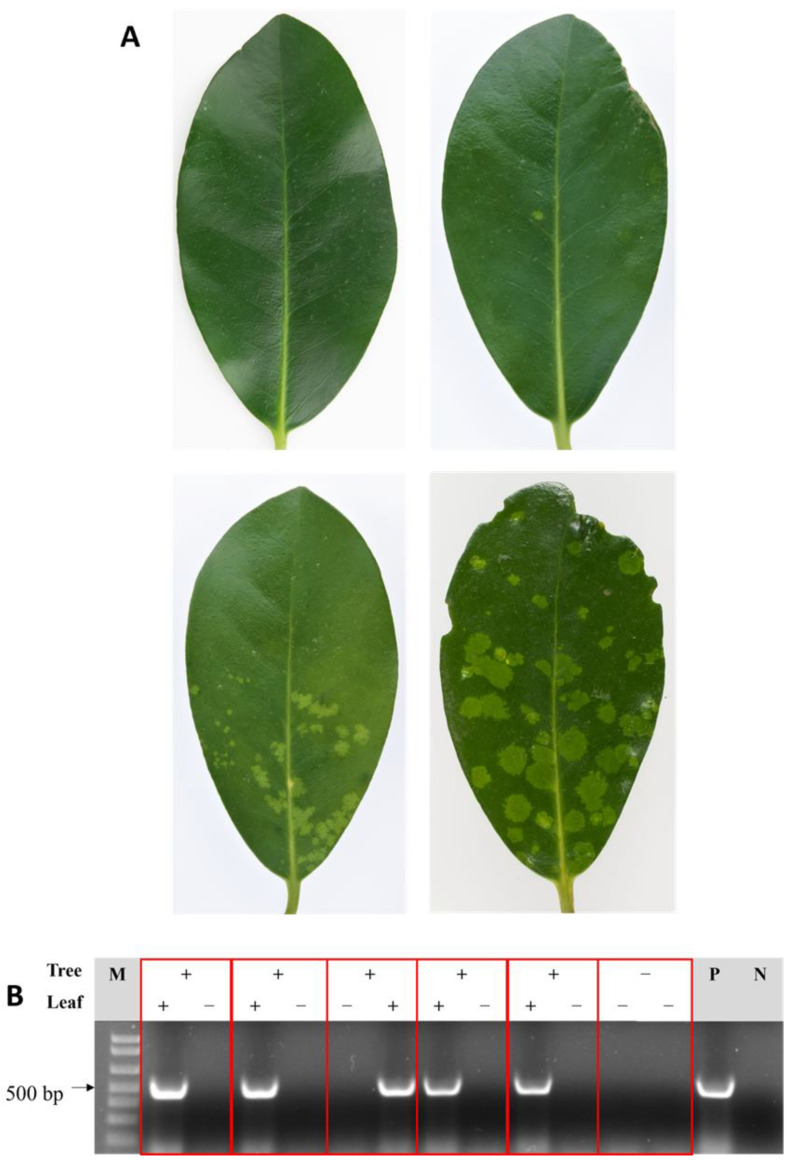
(**A**) Karaka leaves showing no symptoms (top left), or symptoms associated with the presence of Karaka Ōkahu purepure virus: a single small chlorotic spot near the midrib (top right), moderate chlorotic spots (bottom left), and strong symptoms (bottom right). (**B**) Gel displaying the results of the specific RT-PCR from symptomatic (+) and asymptomatic (−) tree bearing either symptomatic (+) and/or asymptomatic (−) leaves. Leaves from the same tree are boxed together. P and N indicate positive (RNA from a known infected tree) and negative (water substituting RNA) controls, respectively.

**Figure 2 viruses-13-01611-f002:**
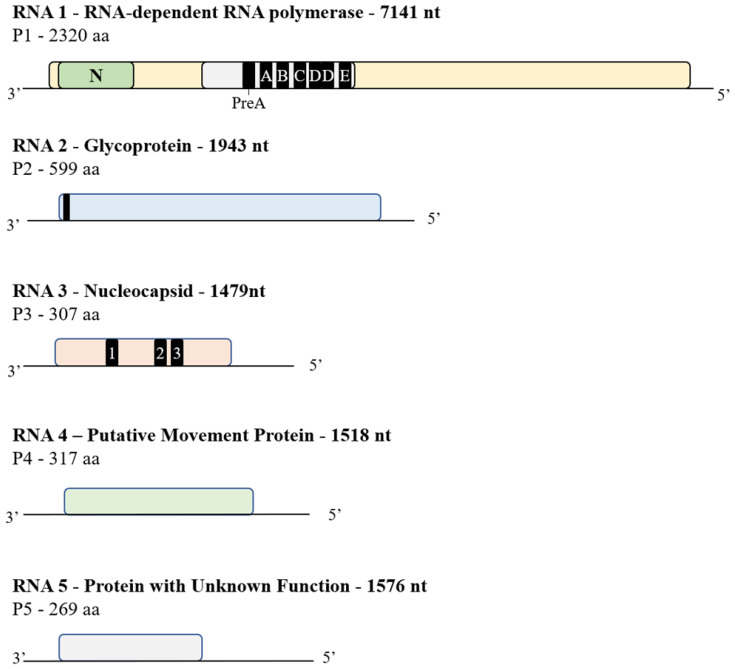
Genomic structure of the five negative-sense RNA segments identified for the KOPV genome. RNA 1: “N” is the putative N-terminus endonuclease domain and Pre-A, A, B, C, DD and E are emaravirus predicted RdRp motifs. RNA 2: black bar represents a putative signal peptide with cleavage site. RNA 3: 1, 2 and 3 represent conserved emaravirus motifs. Note: these segments are not to scale.

**Figure 3 viruses-13-01611-f003:**
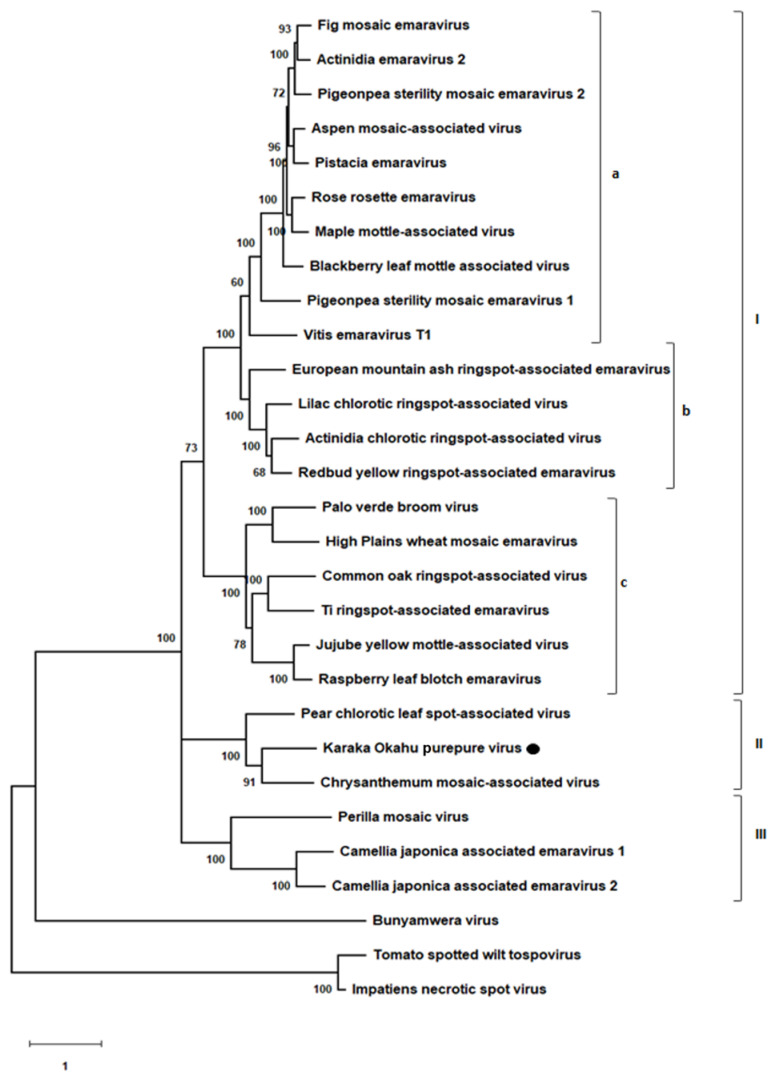
Phylogenetic tree inferred using the amino acid sequences of the putative RdRp. The tree was constructed using the maximum likelihood method with 1000 bootstraps. Sequences used in this analysis are listed in Appendix A. Boostrap values are indicated at the nodes. The scale bar represents the number of amino acid substitutions per site. The three major clades are I, II and III, with the largest clade (clade I) split into three subclades (subclades a, b and c).

**Figure 4 viruses-13-01611-f004:**
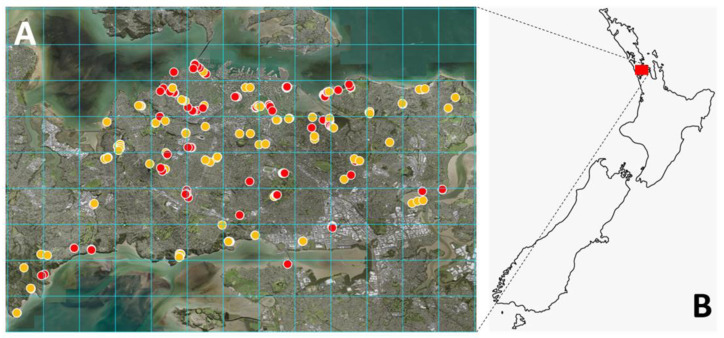
Distribution of Karaka Ōkahu purepure virus in central Auckland as assessed by visual symptoms. (**A**,**B**) Red dots represent symptomatic trees, orange dots represent asymptomatic trees. This map is overlaid with a 2 km × 2 km grid.

**Figure 5 viruses-13-01611-f005:**
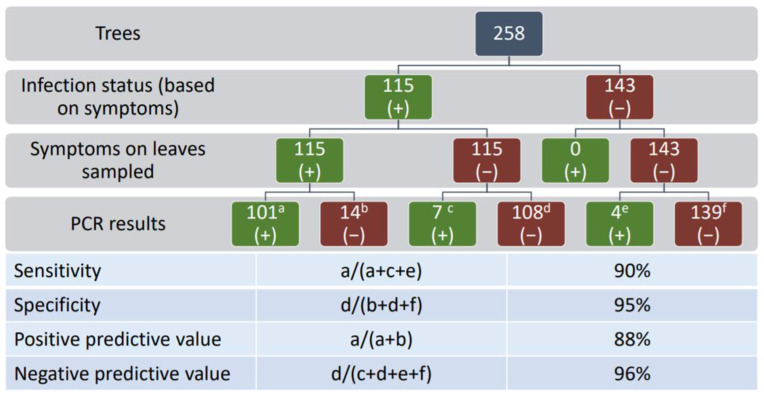
Karaka Ōkahu purepure virus diagnostic test results from 258 trees. Symptomatic and asymptomatic leaves were tested from 115 symptomatic trees and asymptomatic leaves were tested from 143 asymptomatic trees. a RT-PCR positive from a symptomatic leaf; b RT-PCR negative from a symptomatic leaf; c RT-PCR positive from an asymptomatic leaf taken from a symptomatic tree; d RT-PCR negative from an asymptomatic leaf taken from a symptomatic tree; e RT-PCR positive from an asymptomatic leaf taken from an asymptomatic tree; f RT-PCR negative from an asymptomatic leaf taken from an asymptomatic tree. na, not applicable.

## Data Availability

The genome sequence data generated is available at GenBank #MZ391827–MZ391831.

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
