# Peer review of "Characterisation and Distribution of Karaka Ōkahu Purepure Virus—A Novel Emaravirus Likely to Be Endemic to New Zealand"

_viruses, 2021, doi:10.3390/v13081611_

Round 1
Reviewer 1 Report
Rabbidge et al describe the discovery of a new emaravirus in a New Zealand native tree, the karaka. The virus sequence was first obtained using illumine sequence and RACE. The full genome sequence was then analysed, and its features compared with that of other emaraviruses. A survey was the carried out of 258 symptomatic and asymptomatic trees and diagnostics done using RT-PCR the results of which showed the virus to be widespread around the Auckland area.
The manuscript is well written (though a little wordy in places) and makes an interesting story. I would recommend publication assuming the below points can be addressed. At points structurally the paper is confusing as the authors seem to mix results in with materials and methods and the introduction reads as if an emaravirus and its possible vector was already identified before the work was carried out (lines 73 -98).
The statement – “We report the first emaravirus and, to the best of our knowledge, the first endemic plant-associated virus from Aotearoa New Zealand.” is maybe an overstatement. “We report the first endemic plant-associated emaravirus from Aotearoa New Zealand.” might be more appropriate. Fig mosaic virus would probably be the first emaravirus to New Zealand (Minafra et al (2012) Fig tree viruses in New Zealand.) and the endemic native grass has been reported to host multiple exotic viruses (Delmiglio et al 2010 Incidence of cereal and pasture viruses in New Zealand's native grasses). Whether KOPV’s origins can be traced to karaka has not been demonstrated here.
25 – Emaravirus in italics
25 – amino acid homology
34 - Corynocarpus laevigatus in italics
77 - Aculus corynocarpi italics
100 – Mt - in full
108 – KOPV - ICTV doesn’t regulate virus acronyms at the moment but the use of diacritic marks discouraged.
Fig 1 legend – P and N?
159-175 and sections above are more related to the results section
182 – tomato spotted wilt virus
Fig 2 legend – None of the domains functions have been confirmed so “putative” should precede.
287-293 – what kind of protein similarity algorithms were used? Perhaps using PSI or DELTA blast homologs could be identified.
Fig 3 Legend – using complete amino acid sequences of the putative RdRp. What do the numbers at the nodes mean?
440 – the MP is mainly required for cell-to-cell movement not systemic.
440-455 - is too speculative and should be deleted or reduced given the limited information on the biology of the virus.
467-494 – should be deleted. It’s interesting but not pertinent.
Author Response
Reviewer 1
Comments and Suggestions for Authors
Rabbidge et al describe the discovery of a new emaravirus in a New Zealand native tree, the karaka. The virus sequence was first obtained using illumine sequence and RACE. The full genome sequence was then analysed, and its features compared with that of other emaraviruses. A survey was the carried out of 258 symptomatic and asymptomatic trees and diagnostics done using RT-PCR the results of which showed the virus to be widespread around the Auckland area.
The manuscript is well written (though a little wordy in places) and makes an interesting story. I would recommend publication assuming the below points can be addressed. At points structurally the paper is confusing as the authors seem to mix results in with materials and methods and the introduction reads as if an emaravirus and its possible vector was already identified before the work was carried out (lines 73 -98).
Response: No emaravirus had been identified on karaka before this report as stated in lines 74-75 “…while cucumber mosaic virus and ribgrass mosaic virus are the only two viruses reported in karaka before this report…”. By contrast, the potential vector had already been identified as described in the Introduction (lines 76-89) ending with the statement that “It is suspected that the karaka gall mite is the vector of a newly discovered emaravirus in karaka described hereafter.” in lines 88-89. Furthermore, in lines 99-108 we explicitly layout the activity undertaken in this report.
The statement – “We report the first emaravirus and, to the best of our knowledge, the first endemic plant-associated virus from Aotearoa New Zealand.” is maybe an overstatement. “We report the first endemic plant-associated emaravirus from Aotearoa New Zealand.” might be more appropriate. Fig mosaic virus would probably be the first emaravirus to New Zealand (Minafra et al (2012) Fig tree viruses in New Zealand.) and the endemic native grass has been reported to host multiple exotic viruses (Delmiglio et al 2010 Incidence of cereal and pasture viruses in New Zealand's native grasses). Whether KOPV’s origins can be traced to karaka has not been demonstrated here.
Response: We appreciate the insights from the reviewer. We want to express clearly that this is, to the best of our knowledge, the first virus of New Zealand that is both endemic and plant-associated. We stand by our statement as we cannot categorically say that it is endemic. We have changed the abstract to read “We report the first emaravirus on an endemic plant of Aotearoa New Zealand that is, to the best of our knowledge, the country’s first endemic virus characterised in associated with an indigenous plant.”
On lines 99-102 we have included reference to fig mosaic virus and other exotic viruses identified in indigenous plants of the country…” To date, the only emaravirus reported in New Zealand is the exotic fig mosaic virus and other viruses identified in indigenous plants of Aotearoa New Zealand are exotic (Veerakone et al 2015, Blouin et al., 2016, Podolyan et al., 2020).”
The Discussion (lines 356-358) now reads “…and how the current collective knowledge indicates KŌPV as the first description of a likely endemic virus of Aotearoa New Zealand on an endemic plant of the country “
25 – Emaravirus in italics
Corrected
25 – amino acid homology
Corrected including “amino acid” changed and “homology” replaced with “similarity” since homology is not measured by percentage while similarity is measured by percentage.
34 - Corynocarpus laevigatus in italics
Corrected
77 - Aculus corynocarpi italics
Corrected
100 – Mt - in full
Corrected
108 – KOPV - ICTV doesn’t regulate virus acronyms at the moment but the use of diacritic marks discouraged.
We understand that it is difficult for readers unfamiliar with the Māori language but the name of location is spelt with the macron (Ōkahu Bay). We understand that future, non-NZ publications, reporting that virus will not use the macron, however, to be precise, we believe that it is necessary to keep the macron, especially in this special issue on State of the art of virology in New Zealand.
Fig 1 legend – P and N?
The definition of P and N has been included into the Figure 1 legend.
159-175 and sections above are more related to the results section
We agree that the placement of Figure 1 is problematic. We have removed reference to this figure from the methods section and placed it at the beginning of the results section.
We believe the text of this section is methods, as lines 159-175 describe what was used to predict particular types of sites and signal peptides. 182 – tomato spotted wilt virus
Corrected
Fig 2 legend – None of the domains functions have been confirmed so “putative” should precede.
Corrected
287-293 – what kind of protein similarity algorithms were used? Perhaps using PSI or DELTA blast homologs could be identified.
Section 2.1.2. Genome and protein analysis describes the algorithms used to explore potential protein structures. Some of this section has been reworded to include the use of BLASTn and BLASTx algorithms for greater clarity.
Fig 3 Legend – using complete amino acid sequences of the putative RdRp. What do the numbers at the nodes mean?
Completed
440 – the MP is mainly required for cell-to-cell movement not systemic.
While this might be considered the “main” function of the MP, there have been several studies and reviews that show transport across plasmodesmata (mediated by the MP) as being essential for the systemic infection. Examples are listed.
Hong JS, Ju HJ. The Plant Cellular Systems for Plant Virus Movement. Plant Pathol J. 2017;33(3):213-228. doi:10.5423/PPJ.RW.09.2016.0198.
Yu C, Karlin DG, Lu Y, Wright K, Chen J, MacFarlane S. Experimental and bioinformatic evidence that raspberry leaf blotch emaravirus P4 is a movement protein of the 30K superfamily. J Gen Virol. 2013 Sep;94(Pt 9):2117-2128. doi: 10.1099/vir.0.053256-0. Epub 2013 Jun 12. PMID: 23761405.
440-455 - is too speculative and should be deleted or reduced given the limited information on the biology of the virus.
We understand the reviewer’s reservations about this section. However, we would like this section to remain albeit abbreviated. We are proposing a hypothesis and feel it warrants discussion. It outlines avenues for further research and opens these ideas up for debate, which we believe is an essential component of publications.
467-494 – should be deleted. It’s interesting but not pertinent.
We would like this section to remain as is. This outlines important cultural connections for this virus, as well as explores the possibility of whether or not this virus is endemic to NZ. Identifying this virus as being endemic has important cultural implications for all NZers, since all indigenous flora and fauna are claimed by Maori to be under their care and guardianship...

Reviewer 2 Report
In general, the manuscript merits publication because it contains important data for the reader and to the scientific community. However, here are brief comments\suggestions\queries of a general nature to be taken into consideration before publication.
- Unnecessarily long description in some sections; especially in the introduction and in the final discussion, where some concepts already expressed in the results are repeated. So please try to be concise.
- Excessive emphasis on aspects not related to the study of viruses (description of the crop, places where it is grown, intended use, etc., too long).
- unneeded information regarding the determination of the sensitivity and specificity parameters of the visual diagnosis for this virus (part of paragraph 3.3 and all of Figure 5). Does it make sense to talk about the specificity of visual diagnosis? And what does mean to speak of specificity? Specificity compared to what other viruses? For me this paragraph needs to be almost completely removed. It should only be highlighted the high correlation between symptom and presence of the virus, unlike what happens in asymptomatic samples.
- Rather, always remaining on this aspect, the authors do not provide any explanation on the symptomatic but negative samples. Are these symptoms due to other pathogens or adverse factors? Or are they variants of the same virus not detected by the primers used in the survey?
- Similarly, given that the authors go so far as to hypothesize that the lack of the virus is due to the poor functionality of the "Movement protein" which has some differences from the MP of other emaraviruses, and if so, why they do not try to explain why the virus is detected even in asymptomatic samples?
- The diagnosis of the samples remains a great mystery throughout the work. From what I understand, they use the Multiplex (against all 5 viral RNAs) for the diagnosis of the original sample from which the study started. On the other hand, it is not clear whether the Multiplex was always used for subsequent studies or only a simple PCR (to which RNA?). It is not clear from the text. If, as they say, the test was done with the Multiplex, why is only one band visible in figure 1?
- Why did they not try to use the primers constructed on the other RNAs in the negative symptomatic PCR samples?
- Also in the text, in paragraph 2.2.3, the degenerate primers (here called generic rather than universal, line 204) and those of Chooi et al., 2017 are also mentioned, but there is no trace of them in the results. Where and when were they used?
- I would also review the name of the virus: the terms Okabu "" Purepure "are incomprehensible to most. I would propose "Auckland karaka mottling virus", where Auckland indicates a much wider and better known geographical area; I would use the term mottling (or spotted) instead of purepure in the Maori language.
- In the abstract I would say novel virus, rather than new-to-science virus (also present in the Keywords); genus Emaravirus (and non-family emaravirus); I would speak of most conserved RNA1, rather than most conserved gene (RNA1) since RNA is not a gene; I would eliminate “in te reo maori” (line 28, which I don't know what it means);
- In line 103 he speaks again of family members, referring to emaraviruses (I would put genus members).
- In line 163 authors speak of a “manual search” which I have not understood.
- In line 174 "emaraviruses" instead of "emaravirus"
- In the discussion the authors go into proposals which in my opinion are very premature for these species and scenarios given here are all speculative. Regarding, for example, to the subdivision of emaraviruses into groups and subgroups; to the question whether man or natural spread by seeds contributed to spreading the virus is speculative. None of the 22 emaraviruses is reported to be transmitted by seeds. If so, why question it; diffusion through eryophid mites is much more likely. Too much space is then dedicated to the vector, for which absolutely nothing was done in this work. The author should therefore limit their discussion to saying that, as with other emaraviruses, the eryophid mites of this species "could" be natural vectors.
- In addition, what great importance and sense it has to understand where, how and when the virus started: if it is present in the villa of the mayor or vice-mayor or someone else: to conclude what? It seems to me an excess of scientific speculation that leads to absolutely nothing (lines 425-427).
- The bibliography should be rearranged by putting it in alphabetical order (see eg n. 2 and n. 63). Honestly, I have not checked if there is full correspondence between the works cited in the text and those listed in the bibliography.
- In my opinion the work needs a linguistic revision, although this is less important than a revision in the contents.
- There would be many other small things, that I didn’t consider in my revision too.
Author Response
Reviewer 2
Comments and Suggestions for Authors
In general, the manuscript merits publication because it contains important data for the reader and to the scientific community. However, here are brief comments\suggestions\queries of a general nature to be taken into consideration before publication.
Unnecessarily long description in some sections; especially in the introduction and in the final discussion, where some concepts already expressed in the results are repeated. So please try to be concise.
Excessive emphasis on aspects not related to the study of viruses (description of the crop, places where it is grown, intended use, etc., too long).
The authors are very mindful of the potential impact of the findings on the Māori community and have made every effort to respect Māori knowledge, and have therefore consulted with the Māori from Ōkahu, Auckland. There is significant cultural importance of the karaka in NZ, and all indigenous flora and fauna are claimed by Maori to be under their care and guardianship. Māori often convey their information through stories, therefore we have made efforts to incorporate important information. For this reason, we would like these sections to remain as they are.
unneeded information regarding the determination of the sensitivity and specificity parameters of the visual diagnosis for this virus (part of paragraph 3.3 and all of Figure 5). Does it make sense to talk about the specificity of visual diagnosis? And what does mean to speak of specificity? Specificity compared to what other viruses? For me this paragraph needs to be almost completely removed. It should only be highlighted the high correlation between symptom and presence of the virus, unlike what happens in asymptomatic samples.
The comparison is valid.
Rather, always remaining on this aspect, the authors do not provide any explanation on the symptomatic but negative samples. Are these symptoms due to other pathogens or adverse factors? Or are they variants of the same virus not detected by the primers used in the survey?
The range of symptoms is large and maybe some of the different symptoms may be caused by other stresses. There is sufficient data in the paper for others to perform additional research on this aspect.
Similarly, given that the authors go so far as to hypothesize that the lack of the virus is due to the poor functionality of the "Movement protein" which has some differences from the MP of other emaraviruses, and if so, why they do not try to explain why the virus is detected even in asymptomatic samples?
The MP has been addressed for Reviewer 1. Positive detection in asymptomatic samples may be due to nascent infection or non-obvious symptoms.
The diagnosis of the samples remains a great mystery throughout the work. From what I understand, they use the Multiplex (against all 5 viral RNAs) for the diagnosis of the original sample from which the study started. On the other hand, it is not clear whether the Multiplex was always used for subsequent studies or only a simple PCR (to which RNA?). It is not clear from the text. If, as they say, the test was done with the Multiplex, why is only one band visible in figure 1?
We have added text in the M&M to clarify the multiplex diagnostic test which gives a single band even from multiple amplicons.
Why did they not try to use the primers constructed on the other RNAs in the negative symptomatic PCR samples?
All samples (symptomatic and asymptomatic) were tested for KŌPV using the multiplex that targeted all KŌPV RNAs.
Also in the text, in paragraph 2.2.3, the degenerate primers (here called generic rather than universal, line 204) and those of Chooi et al., 2017 are also mentioned, but there is no trace of them in the results. Where and when were they used?
We have modified the relevant M&M text and added two new sentences to the results.
“A subset of RNA samples from 40 asymptomatic leaves from symptomatic karaka trees and 40 asymptomatic leaves from asymptomatic trees were tested with the internal primers VvNAD5 which validated the karaka RNA isolation method and the integrity of RNA for the RT-PCR (data not shown).” (lines 319-322)
Using RNA from symptomatic leaves the multiplex RT-PCR diagnostic test gave stronger amplification than the generic primers of Elbeaino et al (2013, data not shown) therefore the multiplex was used subsequently.” (lines 326-328)…”
I would also review the name of the virus: the terms Okabu "" Purepure "are incomprehensible to most. I would propose "Auckland karaka mottling virus", where Auckland indicates a much wider and better known geographical area; I would use the term mottling (or spotted) instead of purepure in the Maori language.
With respect, there is acceptance of the use of Latinised, alphanumeric and free form names in taxonomy including use of indigenous names (Gillman and Wright 2020) and this has been introduced into virology as exampled by Varsani and Krupovic (2017) and Walker et al. (2021). Naming of this virus has been done in consultation with Māori as a mark of respect since this virus has been found in a native, taonga (sacred) plant species. This is in line with the increasing awareness that indigenous knowledge and inclusion are important in research.
Gillman LN, Wright SD. Restoring indigenous names in taxonomy. Commun Biol. 2020 Oct 23;3(1):609. doi: 10.1038/s42003-020-01344-y. PMID: 33097807; PMCID: PMC7584613.
Walker PJ, Siddell SG, Lefkowitz EJ, Mushegian AR, Adriaenssens EM, Alfenas-Zerbini P, Davison AJ, Dempsey DM, Dutilh BE, García ML, Harrach B, Harrison RL, Hendrickson RC, Junglen S, Knowles NJ, Krupovic M, Kuhn JH, Lambert AJ, Łobocka M, Nibert ML, Oksanen HM, Orton RJ, Robertson DL, Rubino L, Sabanadzovic S, Simmonds P, Smith DB, Suzuki N, Van Dooerslaer K, Vandamme AM, Varsani A, Zerbini FM. Changes to virus taxonomy and to the International Code of Virus Classification and Nomenclature ratified by the International Committee on Taxonomy of Viruses (2021). Arch Virol. 2021 Sep;166(9):2633-2648. doi: 10.1007/s00705-021-05156-1. PMID: 34231026.
Varsani A, Krupovic M. Sequence-based taxonomic framework for the classification of uncultured single-stranded DNA viruses of the family Genomoviridae. Virus Evol. 2017 Feb 2;3(1):vew037. doi: 10.1093/ve/vew037. PMID: 28458911; PMCID: PMC5399927.
In the abstract I would say novel virus, rather than new-to-science virus (also present in the Keywords); With respect we prefer new-to-science over novel.
genus Emaravirus (and non-family emaravirus); Corrected as suggested.
I would speak of most conserved RNA1, rather than most conserved gene (RNA1) since RNA is not a gene; Corrected as suggested.
I would eliminate “in te reo maori” (line 28, which I don't know what it means); We have included the translation so that the phrase now reads “te reo Māori (the Māori language)”
In line 103 he speaks again of family members, referring to emaraviruses (I would put genus members). Corrected as suggested.
In line 163 authors speak of a “manual search” which I have not understood.
This has been removed.
In line 174 "emaraviruses" instead of "emaravirus" Corrected as suggested.
In the discussion the authors go into proposals which in my opinion are very premature for these species and scenarios given here are all speculative. Regarding, for example, to the subdivision of emaraviruses into groups and subgroups; to the question whether man or natural spread by seeds contributed to spreading the virus is speculative. None of the 22 emaraviruses is reported to be transmitted by seeds. If so, why question it; diffusion through eryophid mites is much more likely. Too much space is then dedicated to the vector, for which absolutely nothing was done in this work. The author should therefore limit their discussion to saying that, as with other emaraviruses, the eryophid mites of this species "could" be natural vectors.
To clarify we have added “(perhaps by eriophyid mites)”in line 426 so that it now reads “Another scenario for the even distribution of KŌPV observed in Auckland is that KŌPV initially spread (perhaps by eriophyid mites) through a nursery, followed by an artificial spread by human activity to the rest of Auckland and potentially wider New Zealand through planting of the infected trees.”
In addition, what great importance and sense it has to understand where, how and when the virus started: if it is present in the villa of the mayor or vice-mayor or someone else: to conclude what? It seems to me an excess of scientific speculation that leads to absolutely nothing (lines 425-427).
It is important to Maori and to New Zealand society in general to know whether an organism is indigenous or not. All indigenous flora and fauna are claimed by Maori to be under their care and guardianship. We empathise that the implications are cultural and may not be easy to comprehend for non-New Zealanders however we have specifically elected to submit this manuscript to a special issue on State of the art of virology in New Zealand for these reasons.
The bibliography should be rearranged by putting it in alphabetical order (see eg n. 2 and n. 63). Honestly, I have not checked if there is full correspondence between the works cited in the text and those listed in the bibliography.
We have checked the correspondence between works cited and those listed and have corrected the erroneous order.
In my opinion the work needs a linguistic revision, although this is less important than a revision in the contents.
There would be many other small things, that I didn’t consider in my revision too.
We hope that we have identified other edits that were required.
